# Biosynthesis and Characterization of Calcium Oxide Nanoparticles from *Citrullus colocynthis* Fruit Extracts; Their Biocompatibility and Bioactivities

**DOI:** 10.3390/ma16072768

**Published:** 2023-03-30

**Authors:** Mubsher Mazher, Muhammad Ishtiaq, Bilqeesa Hamid, Shiekh Marifatul Haq, Atiya Mazhar, Faiza Bashir, Mussaddaq Mazhar, Eman A. Mahmoud, Ryan Casini, Abed Alataway, Ahmed Z. Dewidar, Hosam O. Elansary

**Affiliations:** 1Department of Botany, Mirpur University of Science and Technology (MUST), Mirpur 10040, Pakistan; drishiaq.bot@must.edu.pk (M.I.); scholar.faiza@gmail.com (F.B.); mussaddaqbiology60@gmail.com (M.M.); 2Department of Chemistry, University of Kashmir Srinagar, Srinagar 190006, India; bhatbilqees94@gmail.com; 3Department of Ethnobotany, Institute of Botany, Ilia State University, Tbilisi 0162, Georgia; marifat.edu.17@gmail.com; 4Department of Chemistry, Government Post Graduate College for Women, Bhimber 10038, Pakistan; atiyachemistry60@gmail.com; 5Biological Research Center, Institute of Plant Biology, 6726 Szeged, Hungary; 6Department of Food Industries, Faculty of Agriculture, Damietta University, Damietta 34511, Egypt; emanmail2005@yahoo.com; 7School of Public Health, University of California, 2121 Berkeley Way, Berkeley, CA 94704, USA; ryan.casini@berkeley.edu; 8Prince Sultan Bin Abdulaziz International Prize for Water Chair, Prince Sultan Institute for Environmental, Water and Desert Research, King Saud University, Riyadh 11451, Saudi Arabia; aalataway@ksu.edu.sa (A.A.); adewidar@ksu.edu.sa (A.Z.D.); 9Department of Agricultural Engineering, College of Food and Agriculture Sciences, King Saud University, Riyadh 11451, Saudi Arabia; 10Department of Plant Production, College of Food & Agriculture Sciences, King Saud University, Riyadh 11451, Saudi Arabia

**Keywords:** green synthesis, calcium oxide nanoparticles, in vitro release, cytotoxicity, antimicrobial, antioxidant, hemolytic activity

## Abstract

Modern nanotechnology encompasses every field of life. Nowadays, phytochemically fabricated nanoparticles are being widely studied for their bioactivities and biosafety. The present research studied the synthesis, characterization, stability, biocompatibility, and in vitro bioactivities of calcium oxide nanoparticles (CaONPs). The CaONPs were synthesized using *Citrullus colocynthis* ethanolic fruit extracts. Greenly synthesized nanoparticles had an average size of 35.93 ± 2.54 nm and showed an absorbance peak at 325 nm. An absorbance peak in this range depicts the coating of phenolic acids, flavones, flavonols, and flavonoids on the surface of CaONPs. The XRD pattern showed sharp peaks that illustrated the preferred cubic crystalline nature of triturate. A great hindrance to the use of nanoparticles in the field of medicine is their extremely reactive nature. The FTIR analysis of the CaONPs showed a coating of phytochemicals on their surface, due to which they showed great stability. The vibrations present at 3639 cm^−1^ for alcohols or phenols, 2860 cm^−1^ for alkanes, 2487 cm^−1^ for alkynes, 1625 cm^−1^ for amines, and 1434 cm^−1^ for carboxylic acids and aldehydes show adsorption of phytochemicals on the surface of CaONPs. The CaONPs were highly stable over time; however, their stability was slightly disturbed by varying salinity and pH. The dialysis membrane in vitro release analysis revealed consistent nanoparticle release over a 10-h period. The bioactivities of CaONPs, *C. colocynthis* fruit extracts, and their synergistic solution were assessed. Synergistic solutions of both CaONPs and *C. colocynthis* fruit extracts showed great bioactivity and biosafety. The synergistic solution reduced cell viability by only 14.68% and caused only 16% hemolysis. The synergistic solution inhibited *Micrococcus luteus* slightly more effectively than streptomycin, with an activity index of 1.02. It also caused an 83.87% reduction in free radicals.

## 1. Introduction

Nanotechnology is a growing field of material science and research nowadays [1]. The synthesis of metallic nanoparticles from plant-oriented extracts is gaining more interest from scientists due to their cost-effectiveness and rapid action [2]. Nanoparticles (NPs) synthesized from plant resources are of great importance due to their vast applications [3]. In general, nanoparticles are extremely toxic and, due to their smaller size, can easily and freely penetrate biological membranes [4]. Plant secondary metabolites have a great ability to reduce, coat, and stabilize the nanoparticles. Phytochemicals can be coated on the surface of nanoparticles synthesized by the green method [5].

Organically produced metallic nanoparticles exhibit a wide range of bioactivities. *Aloe vera* extract-derived calcium phosphate nanoparticles have excellent antibacterial activity against bacterial and fungal species [6]. The leaf extract of *Piper betel* extract-derived calcium oxide nanoparticles (CaONPs) has strong antibacterial, antioxidant, and antibiofilm properties [7]. *Thymbra spicata* leaf extract-derived biogenic metallic nanoparticles have strong bactericidal activity that is comparable to industry standards. These metallic nanoparticles have an antioxidant activity of about 79.67% [8]. *Moringa olifera* leaf extracts can be used to easily synthesize CaONPs that have powerful antibacterial and antioxidant properties [9]. Calcium oxide nanoparticles produced from *Trigona sp.* show great antifungal potential and pose less toxicity [10].

Metallic nanoparticles show great fabrication by phytochemicals and prove to be less toxic [11]. Calcium apatite nanoparticles show significant apoptosis in some cancer cell lines. The cytotoxic activity displayed by calcium apatite nanoparticles is comparable to the standards [12]. The CaONPs prepared from *Linum usitatissimum* leaf extracts are found to be safer for in vivo treatments [13]. Recently, a study on rats has reported different concentrations of calcium oxide nanoparticles that were administered for 60 days, with no toxicity evidence [14]. Due to the fabrication of phytochemicals on the surface of metallic nanoparticles, they show great biocompatibility [15].

*Citrullus colocynthis* (L.) Schrad. is a medicinal plant that has been used in several traditional medicinal cultures [16]. *C. colocynthis* is rich in cucurbitacins [17], flavonoids [18], phenolics, and natural antioxidants [19]. Plant secondary metabolites are helpful to reduce, cap, and stabilize metallic nanoparticles [20]; phenolics and flavonoids are the most effective biological reducers [21]. Calcium oxide is biocompatible and possess several biomedical applications, specifically antibacterial potential [22]. Green synthesis of calcium oxide nanoparticles from organic molecules is a cost-effective and easy method [23].

This study presents a simple, ecofriendly, cost-effective, and easy method for the synthesis of calcium oxide nanoparticles (CaONPs) from *C. colocynthis* fruit extracts (CCFE) for the first time. It will elaborate on the physical and chemical nature of greenly synthesized CaONPs by UV-Vis spectroscopy, transmission electron microscopy (TEM), X-ray diffraction (XRD), and Fourier-transform infrared spectroscopy (FTIR). It will investigate in vitro–in vivo correlation (IVIVC) and the stability of greenly synthesized CaONPs for the first time. Greenly synthesized nanoparticles will be studied for biosafety by studying hemolytic activity and cytotoxicity assessment against macrophages. The antibacterial potential of CaONPs, CCFE, and their synergistic solution will be investigated against skin-borne pathogens, which has not been achieved before. Antioxidant potential will also be studied using different assays. Findings of this study will suggest the suitability and biocompatibility of greenly synthesized CaONPs for in vitro and in vivo bioactivities.

## 2. Materials and Methods

### 2.1. Ethics Declaration

Three Wister albino male rats (*Rattus norvegicus*) were kept according to the departmental ethics committee (DEC) of Mirpur University of Science and Technology (MUST), Azad Jammu and Kashmir (AJ&K), Pakistan, in line with international standards (http://ors.ubc.ca/contents/animal-care-sops-guidelines accessed on 25 July 2021). Prior to the experiments, permission was obtained from DEC via letter 730/DEC/BOT/2021 dated 25 July 2021. Strict monitoring according to international standard operating procedures and guidelines for experimental organisms was conducted. Test organisms were kept in the laboratory for 10 days prior to the trials. Test organisms were fed abundant grains and water. The temperature was adjusted to 25 °C and the humidity was manually maintained at 60 ± 10%. Dark and light durations were artificially adjusted to 12/12 h. During the experimental period, rats were given access to abundant water and libitum feeding.

### 2.2. Collection of Plants

Approval for plant collection was taken from the departmental plants conservation committee (PCC) under letter no. 519/PCC/BOT/2021, dated 25 July 2021. Plants were collected by Mubsher Mazher, a Ph. D. student at MUST, Mirpur, AJ&K, Pakistan, from wild localities in Dhander, district Bhimber, AJ&K. Geographically, it is located at 32°28′0″ N latitude and 75°6′0″ E longitude. Plants were identified by Dr. Muhammad Ishtiaq (Ph.D.), Professor, Department of Botany, MUST, AJ&K, Pakistan. An identified herbarium, no. MUST BOT. MUH-517, was submitted to the MUST University Herbarium (MUH) of the Botany Department, MUST, AJ&K, Pakistan, for reference. Completely healthy and fresh plants were uprooted, and the soil was discarded. Fruits were separated and shade-dried for two months, and dried fruits were ground into a fine powder.

### 2.3. Extraction and Phytochemical Analysis

The cold soaking method was used for crude extraction, following Mazher et al. [24]. A 20-g fruit powder was soaked in 100 mL of ethanol (Merck, Darmstadt, Germany). After soaking for 7 days, the solution was filtered with Whatman filter paper (No. 42) and dried with the help of a rotary evaporator (Eyela N1100, Shanghai, China). The sticky, dried residue was saved for future experiments.

The confirmation of alkaloids, flavonoids, glycosides, saponins, tannins, and terpenoids was performed following Harborne [25]. For confirmation of alkaloids, Dragendrof, Mayer, Wagner, and Hager tests were performed. Flavonoids were confirmed using alkaline reagent, ferric chloride, and lead acetate tests. A bromine water test and foam test were performed for confirmation of saponins. For confirmation of tannins, a ferric chloride test was performed. Terpenoids were confirmed by the Salkowski tests. The ferric chloride test was used to determine phenols, as described by Harith et al. [26].

### 2.4. Green Synthesis and Characterization of Calcium Oxide Nanoparticles

To greenly synthesize CaONPs, the previously reported method of Ramli et al. [27] was followed, and 50 mL of 0.2 M CaCl_2_.2H_2_O (Merck, Germany) solution was added to 50 mL of *C. colocynthis* fruit extracts. To keep the pH at 10.5, 10 mL of 2.0 M aqueous NH_4_OH (Merck, Germany) was added dropwise and agitated for 30 min, until a white, milky precipitation of Ca(OH)_2_ was visible. The resultant mixture was centrifuged for 15 min at 10,000 rpm (SIGMA^®^ 1–14, Roedermark, Germany). To eliminate any unreacted starting material, the product was rinsed repeatedly with distilled water. The white Ca(OH)_2_ particles were then calcinated at 700 °C for three hours. A white powder was obtained, which was CaONPs. It was characterized by UV-Vis spectroscopy, transmission electron microscopy (TEM), X-ray diffraction (XRD), and Fourier-transform infrared spectroscopy (FTIR).

To characterize CaONPs, previously reported methods by Ahmad et al. [28], Khine et al. [29], and Marquis et al. [30] were followed. To study the optical properties of CaONPs, the colloidal solutions were studied by a UV-Vis spectrophotometer (UV-1900i, Shimadzu, Kyoto, Japan), and absorbance was adjusted between 250 and 500 nm. The speed of scanning was adjusted to 200 nm/min. To study size, shape, and elemental properties, a transmission electron microscopy (TEM) micrograph was taken using a transmission electron microscope (JEM-ARMTM 200 JEOL, Tokyo, Japan). To study the structural properties of CaONPs, X-ray diffraction (XRD) analysis was performed using a diffractometer (JEOL-JSX-3201M, Kyoto, Japan). The drop casting method was adopted, and conditions for XRD were a sampling speed of 2θ/min, λ = 0.15418 nm, a Cu K_α_ radiation source of 40 kV and 30 mA, and 2θ of 20–70°. To confirm the coating of CaONPs with phytochemicals, FTIR spectroscopy was performed. Vibrational properties were studied by an FTIR atomic absorption spectroscope (IRAffinity-1S MIRacle 10, Shimadzu, Kyoto, Japan) with KBr pellets. Measurements were carried out at room temperature, and the wavenumber was 400–4000 cm^−1^.

### 2.5. Stability Analysis of Calcium Oxide Nanoparticles

Stability analysis of nanoparticles was performed following Leonard et al. [31]. To study the stability of calcium oxide nanoparticles in different saline environments, 1 mg of phytochemical-capped nanoparticles was added separately into 1 mL of a 0.2 M aqueous solution of 5% NaCl at different concentrations (5 µL/mL, 10 µL/mL, 15 µL/mL, and 20 µL/mL). These solutions were incubated at 37 °C for 30 min. After incubation for 30 min, absorbance in the range of 250 to 500 nm was recorded. The stability of nanoparticles was also checked at different pH ranges. Phytochemical-capped nanoparticles (1 mg) were added to 1 mL of 1 mM HCl pH 3.1 (Merck, Darmstadt, Germany), 1 mM H_2_CO_3_ pH 4.68 (Merck, Darmstadt, Germany), 1 mM NaHCO_3_ pH 8.27 (Merck, Darmstadt, Germany), and 1 mM NaOH pH 10.98 (Merck, Darmstadt, Germany). These solutions were incubated at 37 °C for 30 min. After incubation for 30 min, absorbance in the range from 250 to 500 nm was recorded. The effect of time on the stability of phytochemical-capped nanoparticles was also assessed. After 5, 10, 20, and 30 days at 4 °C, a solution was prepared by dissolving 1 mg of phytochemical-capped nanoparticles stored at 25 °C in 1 mL of distilled water, and the absorbance of this solution was measured in the range of 250 to 500 nm.

### 2.6. In Vitro Release of Calcium Oxide Nanoparticles

To study the in vitro release of CaONPs, the dynamic dialysis method previously described by Peng et al. [32] was followed. Dialysis membrane (8 kDa) (Biolab, Islamabad, Pakistan) and phosphate buffered saline (PBS) with pH 7.4 (Biolab, Islamabad, Pakistan) were generously gifted by the Chinaar free dialysis center, Bhimber, AJ&K, Pakistan. Concisely, 0.5, 1, 2, and 4 mg of CaONPs, dissolved in 1 mL of ethanol, were packed in a dialysis membrane, and the membrane was carefully bound tightly by both ends. A release medium was prepared by dissolving 0.5 mL of Tween^®^ 80 (Merck, Darmstadt, Germany) in 100 mL of PBS. A dialysis bag containing CaONPs was dipped in 10 mL of release medium, and this beaker was kept in a horizontal shaker (Shimdzu, Kyoto, Japan) for 1 h at 37 °C. After 1 h, the dialysis bag was kept out of the release medium and was centrifuged at 8000 rpm for 3 min. After centrifugation, the supernatant was collected in a clean cuvette, and absorbance was noted at 325 nm by UV spectrophotometer. A new release medium (10 mL) was filled in the beaker, and a dialysis bag containing nanoparticles was again dipped in it and placed in the shaker. The whole process was repeated 10 times.

### 2.7. Determination of Cytotoxic Activity

The cytotoxicity of test solutions was determined following Mosmann [33] colorimetric 3-(4,5-Dimethylthiazol-2-yl)-2,5-diphenyl-tetrazoli-umbromide (MTT) assay and the water-soluble tetrazolium (WST-8) assay. Three test solutions were prepared including *C. colocynthis* fruit extracts solution (CCFE) containing 1 mg of dried fruit extracts in 1 mL of ethanol, calcium oxide nanoparticles solution containing 1 mg of CaONPs in 1 mL of ethanol, and synergistic solution (SynS) containing 0.5 mg of CCFE and 0.5 mg of CaONPs in 1 mL of ethanol.

#### 2.7.1. Separation of Peritoneal Macrophages

To separate peritoneal macrophages, the Ray and Dittel [34] method was followed. Briefly, 5 mL of ice-cold phosphate buffered saline (PBS) (Biolab, Islamabad, Pakistan) along with 3% fetal calf serum (FCS) (Capricorn Scientific, Ebsdorfergrund, Germany) was injected in the peritoneal cavity of a rat. A gentle massage of the peritoneum was performed to obtain all the available cells from the peritoneal cavity. After the massage, the injected solution was collected back into the syringe. The collected solution containing cell suspensions was centrifuged for 10 min at 1500 rpm. The resulting supernatant was discarded, and one portion of the remaining cells was poured into PBS for counting; the other portion of the cells was poured into a sterile test tube. The viability of the isolated macrophages was tested using trypan blue (Merck, Darmstadt, Germany). The blue-colored cells were designated as dead.

#### 2.7.2. Maintaining In Vitro Culture of Peritoneal Macrophages

To prepare cultures of isolated peritoneal macrophages, the Kumar et al. [35] method was followed. In vitro culture was prepared in Dulbecco’s modified Eagle medium (DMEM) (Sigma-Aldrich, Taufkirchen, Germany), 10% FCS, and 1% streptomycin (Sigma-Aldrich, Taufkirchen, Germany). On a culture plate, 1 mL of isolated peritoneal macrophages was poured. This plate was placed in a CO_2_ incubator (IRMECO GmbH & Co., Lutjensee, Germany) for 2 h at 5% CO_2_ and 37 °C. After an incubation of 2 h, cells that were not adhered to the bottom of the culture plates were washed off with the help of PBS. Ice-cold DMEM was used to collect the adherent cells. The collected cells were counted with the help of a phase contract microscope (CX41 Shimadzu, Tokyo, Japan). For studying cytotoxic activity of CaONPs, the number of macrophages was adjusted to 1.0 × 10^4^ per mL. Quantified macrophages were cultured in DMEM extension medium containing FCS and streptomycin. The culture plates were incubated for 24 h at 5% CO_2_ and 37 °C for further experimentation.

#### 2.7.3. MTT Cytotoxicity Assay

A stock solution of MTT was prepared by dissolving 5 mg/mL of PBS, and it was stored in a dark cabin. Treatment solutions of different concentrations were also prepared by dissolving 0.5, 1, 2, and 4 mg/mL of solutions in PBS. In a 96-well plate, macrophages cultured in DMEM extension medium (DMEM, 10% FCS, and 1% streptomycin) were seeded at a concentration of 1.0 × 10^4^ macrophages per well. This culture plate was kept incubating for 24 h, at 37 °C and 5% CO_2_. After incubation, a line of control wells was established in which no treatment drug was added. In the treatment wells, 10 µL of test solutions was added. Each well received 10 µL of MTT from the stock solution. The culture plate was incubated in a humidified environment at 37 °C and 5% CO_2_ for 4 h. After the incubation of 4 h, the culture medium was carefully removed from the wells, and in each well, 100 µL dimethyl sulfoxide (DMSO) (Merck, Darmstadt, Germany) was added. After vigorously shaking the plate for 5 min, the absorbance of formazan dye was measured at 570 nm using a micropipette reader. The following formula was followed for counting percentage cytotoxicity:Cell viability (%) = (A_t_/A_c_) × 100

Here,

A_t_ = Absorbance by test well;A_c_ = Absorbance by control.

#### 2.7.4. WST-8 Cytotoxicity Assay

The WST-8 assay was conducted to analyze the cytotoxicity of test solutions following the protocols determined by the manufacturer. A stock solution of WST-8 (Cayman Chemicals, Ann Arbor, MI, USA) was prepared by dissolving 5 mg/mL in PBS. Briefly, macrophages cultured in DMEM extension medium (DMEM, 10% FCS, and 1% streptomycin) at a concentration of 1.0 × 10^4^ were seeded in each well of a 96-well plate. This culture plate was kept incubating for 24 h, at 37 °C and 5% CO_2_. After incubation, a line of control wells was established in which no treatment drug was added. In treatment wells, 10 µL of 0.5, 1, 2, and 4 mg/mL test solutions was added. In each well, 10 µL of WST-8 was added, and the culture plate was again incubated at 37 °C and 5% CO_2_ for 4 h. After the incubation, a culture plate was studied for absorbance at 450 nm. The following formula was followed for counting the percentage of cytotoxicity:Cell viability (%) = (A_s_/A_c_) × 100

Here,

A_s_ = Absorbance by sample;A_c_ = Absorbance by control.

### 2.8. Determination of In Vitro Hemolytic Activity

The hemolytic activity of test solutions was assessed by following the published method of Oves et al. [36]. The hemolysis caused by test solutions at different concentrations of 0.5, 1, 2, and 4 mg/mL was measured and compared with the standard hemolytic solution Triton^®^ X-100 (Sigma-Aldrich, Taufkirchen, Germany). Briefly, 10 mL of blood was obtained from the healthy rats in standard vials with anticoagulant. The blood was centrifuged for 10 min at 3000 rpm. Blood plasma as well as the buffy coat were removed, and the remaining erythrocytes were diluted up to 50% hematocrit using PBS. From this hematocrit solution, 2 mL was poured into 4 different test tubes. In each test tube, 1 mL of test solution was poured. The resulting solutions were incubated for 1 h at 37 °C. After the incubation, solutions were centrifuged for 1 h minutes at 3000 rpm. After the incubation, the absorbance of resulting hemoglobin caused by bursting of erythrocytes was studied at 576 nm. To assess the percentage of hemolysis, the following formula was followed:Hemolysis (%) = (A_t_ − A_c_)/(100 − A_c_) × 100

Here,

A_t_ is the absorbance shown by the treatments;A_c_ is the absorbance shown by the control (PBS).

### 2.9. Determination of Antimicrobial Activity

To determine the microbial zones of inhibition, the agar well diffusion method described by Perez et al. [37] was followed.

#### 2.9.1. Microbial Organisms and Culture Medium

Three identified microbial strains, *Micrococcus luteus* (gram-positive), *Vibrio cholerae,* and *Vibrio parahaemolyticus* (gram-negative), were obtained from the microbiology lab of Dr. Tanveer Hussain (Ph. D.), Assistant Professor, Department of Botany, MUST, Mirpur (AJ&K), Pakistan.

For culturing microbial strains, potato dextrose agar (PDA) medium with pH 5.6 was established following Mazher et al. [38]. First, 39 g of PDA powder (Biolab, Budapest, Hungary) was dissolved in 1 L of distilled water. The mixture was then boiled on a tripod stand and in a spirit lamp until it turned yellowish. All glassware, including petri dishes, micropipettes, cork borer, and glass streak, was autoclaved at 121 °C for 15 min.

#### 2.9.2. Measurement of Inhibition Zones and Activity Index

The zones of inhibition were measured to assess the antimicrobial potential of CaONPs, fruit extracts of *C. colocynthis*, and their synergistic solution. To prevent contaminations, all the procedures were carried out in a laminar flow (ESCO^®^ Changi, Singapore). In all petri dishes, about 10 mL of PDA solution was poured and allowed to solidify for 2 h. After the PDA was solidified, a cork borer of 5 mm was used to dig the wells. About 2 mL of standard control and 2 mL of test solutions at different concentrations were poured into separate wells of the petri plates. The streak method was used to inoculate the microbial cultures. After the inoculation, all the plates were left in an incubator at 37 °C for 48 h and then the zones of inhibition (a clear area with no microbial growth) were measured. All the experiments were set up in triplicate.

To measure the activity index of CaONPs, the following formula described by Maqbool et al. [39] was followed:Activity Index (AI) = ZI (Sample)/ZI (Standard)

Here,

ZI (Sample) = Clear zone shown by the sample;ZI (Standard) = Clear zone shown by standard drug.

### 2.10. Determination of Antioxidant Activity

To determine the antioxidant activity of CaONPs, fruit extracts of *C. colocynthis,* and synergistic solution of both CaONPs and *C. colocynthis* extracts, the published procedure of Brand-Williams et al. [40] was adopted. The 1,1-diphenyl-picrylhydrazine (DPPH) radical scavenging assay (DSRA) was adopted for determining antioxidant activity. Antioxidant activity of CaONPs was compared with that of the standard antioxidant butylated hydroxytoulene (BHT) (Merck, Darmstadt, Germany).

#### 2.10.1. Establishing Test Solutions and Dilutions

From the prepared solutions of different standards, CaONPs, fruit extracts of *C. colocynthis*, and the synergistic solution of both CaONPs and *C. colocynthis* extracts, serial dilutions were prepared. Dilutions of 250, 500, 1000, and 2000 µL/mL were established. For analyzing the antioxidant potential by DSRA, 33 mL of a 0.01 mM DPPH solution was dissolved in 1 L of ethanol (Merck, Darmstadt, Germany).

#### 2.10.2. DPPH Radical Scavenging Assay (DSRA)

To carry out DSRA for each solution, 1 mL of a test solution was taken from the prepared dilutions, and 5 mL of DPPH solution was taken from the stock solution and poured into glass cuvettes. After leaving the cuvettes at room temperature for 30 min, the absorbance at 515 nm was measured. Free radical scavenging activity was determined as the percent inhibition, and the results were analyzed by comparing them with the inhibition shown by the standards. To calculate percent inhibition, the following formula was followed:Inhibition (%) = (A_c_ − A_s_)/A_b_ × 100

Here,

A_c_ is absorbance by the control;A_s_ is absorbance by the sample;A_b_ is absorbance by the blank.

### 2.11. Statistical Evaluation

Arithmetic mean, standard deviation, and post-hoc Duncan multiple range test (DMRT) were analyzed by the Statistical Package for Social Sciences SPSS 16.0 (IBM^®^, New York, NY, USA). All the results are expressed in mean and standard error of means (mean ± SEM). All values of *p* < 0.05 were deliberated as statistically significant. A one-way analysis of variance (ANOVA) was performed to compare the group means. Where the difference was significant, DMRT was carried out.

## 3. Results

### 3.1. Phytochemical Analysis of C. colocynthis Fruit Extract

Basic phytochemicals were investigated using a phytochemical examination of ethanolic extracts of *C. colocynthis* fruit extracts. Alkaloids, phenols, and terpenoids were found in large amounts (Table 1). Glycosides and saponins were just mildly prevalent. Tannins and flavonoids were only found in trace amounts.

### 3.2. Green Synthesis and Characterization of CaONPs

The mixture of calcium chloride and *C. colocynthis* fruit extracts appeared to be a milky solution. The milky solution of CaONPs after centrifugation, calcination, and drying appeared to be a fine powder; its color was white. The greenly synthesized CaONPs were characterized by UV-Vis spectroscopy, TEM, XRD, and FTIR.

#### 3.2.1. UV-Vis Spectroscopy

The UV-Vis spectrograph of calcium oxide nanoparticles showed an absorbance peak at 325 nm (Figure 1).

#### 3.2.2. Transmission Electron Microscopy (TEM)

The transmission electron micrograph (TEM) showed the spherical shape of CaONPs. The size of CaONPs measured by ImageJ (version 1.49i, 2014) software ranged from 32 to 43 nm, with an average of 35.93 ± 2.54 nm (Figure 2).

#### 3.2.3. X-ray Diffraction (XRD)

The X-ray diffraction (XRD) pattern of CaONPs showed sharp peaks, which depicted the crystalline properties of triturate. The diffractive peaks were shown at 29.61° (011), 32.17° (111), 37.27° (200), and 54.26° (022), corresponding to CaO (Figure 3).

#### 3.2.4. Vibrational Properties by FTIR

*C. colocynthis* fruit extract contains alkaloids, flavonoids, glycosides, phenols, saponins, tannins, and terpenoids. These phytochemicals act as stabilizing and capping agents for the conversion of reactive metallic ions of calcium (Ca^2+^) into biocompatible CaONPs. For the confirmation of capping by phytochemicals, greenly synthesized nanoparticles were subjected to FTIR analysis. The vibrational properties of CaONPs’ crystal structures elaborated that some phytochemicals were adsorbed on the surface of CaONPs. The vibrations at 3639 cm^−1^ for -OH, 2860 cm^−1^ for -C-H, 2487 cm^−1^ for -C≡C-, 1625 cm^−1^ for -N-H, 1434 cm^−1^ for -COOH, 1053 cm^−1^ for -C-F, and 530 cm^−1^ for Ca-O show the presence of alcohols or phenols, alkanes, alkynes, amines, carboxylic acids, aldehydes, and alkyl halides adsorbed on the surface of CaONPs (Figure 4). It was noticed that the vibration for the presence of alkanes was intensified in SynS and a new vibration of esters at 1741 cm^−1^. However, the strong vibrations for flavonoids, phenolics, and carboxylic acids were present in all test solutions including CCFE, CaONPs, and SynS.

### 3.3. Assessment of Stability of CaONPs

The stability of CaONPs was studied against salinity, time, and pH variations. The stability of prepared nanoparticles was assessed against different solutions of varying salinity ranging from 5 µL/mL to 20 µL/mL of 0.1 mM NaCl. It was noticed that minute changes occurred in the stability of CaONPs with increasing salinity (Figure 5A). The stability of CaONPs against different solutions of varying pH was also assessed. The stability of CaONPs was found to be significantly affected by pH changes (Figure 5B). The highest deviation from reference was noticed with NaOH (pH 10.98). To assess the stability of CaONPs’ optical density, prepared nanoparticles were assessed every 5 days. CaONPs were found to be highly stable over time (Figure 5C).

### 3.4. Assessment of In Vitro Release of CaONPs

In vitro release of CaONPs was studied by the dynamic dialysis method using a dialysis membrane (8 kDa). It was found that greenly synthesized CaONPs have been reduced to a stable configuration. CaONPs in vitro release was measured every hour for a total of 10 h. A steady concentration of nanoparticles was released with very few variations (Figure 6). It was discovered that an increase in concentration during nanoparticle treatment was directly related to an increase in in vitro release. The released concentration of CaONPs kept increasing from 1 h to 6 h, then an almost steady decrease was noticed up to 10 h. No sharp variations in the release of CaONPs were noticed.

### 3.5. Assessment of Cytotoxic Activity

The in vitro cytotoxicity of CaONPs, *C. colocynthis* fruit extracts, and their synergistic solution were studied by MTT and WST-8 assays, and the results are presented in Figure 7. It was found that all the solutions showed a reduction in cell viability by MTT assay. The cell viability percentages by MTT assay were noticed to be (85.32 ± 0.75 ^c^) by synergistic solution, (76.84 ± 1.05 ^b^) by *C. colocynthis* fruit extracts, (60.67 ± 0.75 ^a^) by CaONPs, and (99.40 ± 0.19 ^d^) by the control. The cell viability determined by the WST-8 assay was significantly reduced by the synergistic solution (85.60 ± 0.98 ^c^), *C. colocynthis* fruit extracts (77.06 ± 0.26 ^b^), and CaONPs (67.34 ± 1.64 ^a^) as compared to the control (99.34 ± 0.38 ^d^) (Appendix A).

### 3.6. Assessment of In Vitro Hemolytic Activity

The hemolytic activity of CaONPs, *C. colocynthis* fruit extracts, and their synergistic solution were assessed and compared with that of triton X-100. Results of the hemolytic analysis are presented in Figure 8. The percentage of hemolysis shown by CaONPs, *C. colocynthis* fruit extracts, and their synergistic solution was 28.45 ± 2.28 ^c^, 10.19 ± 0.99 ^a^, and 16.00 ± 1.61 ^b^, respectively. The percentage of hemolysis shown by different treatments was significantly lower as compared to the standard triton X-100 (97.79 ± 1.52 ^d^) (Appendix A).

### 3.7. Assessment of Antimicrobial Activity

The antimicrobial activity of CaONPs, *C. colocynthis* fruit extracts, and their synergistic solution was assessed by the disc diffusion method and results are presented in Figure 9. The synergistic solution of CaONPs and *C. colocynthis* fruit extracts showed significant antimicrobial activity against all three microbes. The zone of inhibition shown by the synergistic solution against *M. luteus* was (25.6 ± 1.1 ^d^), slightly greater than that of streptomycin (24.8 ± 1.6 ^c^), with an activity index of 1.03 (Table 2).

### 3.8. Assessment of Antioxidant Activity

The antioxidant potential of CaONPs, *C. colocynthis* fruit extracts, and their synergistic solution was assessed by a DPPH radical scavenging assay. Antioxidant activity was found to be dose dependent. There was a significant difference among all the treatment groups as well as all the dilutions studied for their antioxidant activity (Table 3). There was a significant difference between the antioxidant potential of BHT (standard) and all the treatment groups. The synergistic solution demonstrated the highest antioxidant activity (83.87 ± 1.68 ^c^) at a dilution of 2000 µL, which was significantly different as compared to BHT (89.42 ± 4.19 ^d^).

## 4. Discussion

A rapidly expanding area of nanotechnology is the facile manufacturing of metallic nanoparticles. The environmentally friendly production of metallic nanoparticles is a practical method with several advantages. The present research was focused on greenly synthesizing calcium oxide nanoparticles (CaONPs) using *C. colocynthis* fruit extracts. Prepared CaONPs were characterized through UV-Vis spectroscopy, transmission electron microscopy (TEM), and X-ray diffraction (XRD). Prepared CaONPs were characterized through UV-Vis spectroscopy, transmission electron microscopy (TEM), and X-ray diffraction (XRD). A UV-Vis spectrograph of CaONPs synthesized from *C. colocynthis* fruit extracts showed an absorbance peak at 325 nm. Absorbance peak in this range depicts the coating of phenolics [41], flavones, flavonols, and flavonoids [42] on the surface of CaONPs. Adsorption of phytochemicals can be attributed to the calcium ions being reduced to CaONPs. It was evident from the change in color from transparent white to milky white. Results of our research are the same as those reported for 332 nm by Gandhi et al. [43] and broad peaks in the range of 265 to 350 nm by Ramola et al. [44]. A TEM micrograph showed a cubic to elliptical shape of CaONPs and an average size of 35.93 ± 2.54 nm. Previously, CaONPs with average sizes ranging from 29 to 38 nm have been reported by Hussein et al. [45] and 13 to 49 nm by Mostafa et al. [46]. The XRD pattern of CaONPs depicted sharp diffractive peaks that showed the crystalline properties of triturate. The diffractive peaks were shown at 29.61° (011), 32.17° (111), 37.27° (200), and 54.26° (022), corresponding to CaO depicting the preferred cubic crystalline nature and average size of 32.12 nm calculated by following the Scherrer equation. ImageJ (version 1.49i, 2014) software calculated nanoparticle sizes from TEM ranging from 32 to 44 nm, with an average size of 35.93 nm. The XRD peaks were well matched with Joint Committee for Powder Diffraction Standard (JCPDS) Card No. 00-004-0777, also reported by Jadhav et al. [9].

The vibrations at 3639 cm^−1^ for -OH, 2860 cm^−1^ for -C-H, 2487 cm^−1^ for -C≡C-, 1625 cm^−1^ for -N-H, and 1434 cm^−1^ for -COOH show the presence of free alcohols or phenols, alkanes, alkynes, amines, carboxylic acids, and aldehydes adsorbed on the surface of CaONPs. It was noticed that the vibration for presence of alkanes was intensified in SynS and a new vibration of esters at 1741 cm^−1^ was present. However, the strong vibrations for flavonoids, phenolics, and carboxylic acids were present in all test solutions including CCFE, CaONPs, and SynS. These findings are in accordance with previous FTIR studies of greenly synthesized CaONPs by Ahmad et al. [28] and Mostafa et al. [46]. The adsorption of phytochemicals is necessary for metallic nanoparticles to be used as pharmaceuticals. Further investigation into the stability of CaONPs using Plasmon resonance (max) revealed that phytochemical-capped nanoparticles were extremely stable against salinity, time, and pH variations. Our findings are consistent with previously reported results by Leonard et al. [31], and they also reported high stability of nanoparticles coated with ginseng. The stability tests proved that CaONPs were highly stable at varying salinities, pH, and time.

Nanoparticles are being widely studied for their bioactivities, but their ability to form coagulants and extremely reactive nature make them unsuitable for use in living organisms. *C. colocynthis* fruits are traditionally used for antidiabetic activity; modern research has also validated their use in antidiabetic activity due to the presence of phenolics, flavonoids, and cucurbitacins in their extracts [47,48]. Calcium oxide nanoparticles have a great capacity for catalysis and drug delivery [49,50]. Green synthesis of CaONPs using fruit extracts of *C. colocynthis* can be useful for producing nanoparticles capped and stabilized with these phytochemicals and having better bioavailability for treating different ailments including diabetes. Using a dialysis membrane to study in vitro release kinetics is an easy and accurate method to predict in vitro–in vivo correlation (IVIVC) [51]. We used metformin (Neodipar^®^, Sanofi, Pakistan) as a control drug in our in vitro release model. We studied the in vitro release kinetics to depict that these nanoparticles have great bioavailability for up to 10 h. The current study examined the release kinetics of greenly synthesized nanoparticles to determine their suitability for in vivo use. In vitro release kinetics showed a steady-state release of CaONPs, and no sharp changes in release concentrations were noticed, showing that the fabrication of phytochemicals from the fruit extracts of *C. colocynthis* has greatly stabilized the nanoparticles. Similar results for greenly synthesized metallic nanoparticles have been documented by previous research by Abu-Huwaij et al. [52] and Yesil-Celiktas et al. [53]. Findings of our research depict CaONPs synthesized using *C. colocynthis* fruit extracts can be suitable candidates for studying in vivo antidiabetic activity in future.

Due to their nanosize, ability to form coagulations, and extremely reactive nature, nanoparticles can destroy cell membranes, which is a hindrance to their use in living organisms. We have studied the cytotoxic effect of greenly synthesized CaONPs against macrophages to assess their toxicity for living cells. Using macrophages to assess the cytotoxicity of plant extracts is an easy, cost-effective, and reliable method [54,55]. The intention behind studying the biosafety of CaONPs is that, in the future, we want to study in vivo antidiabetic and antilipidemic activity using these nanoparticles. The cytotoxicity analysis showed that greenly synthesized CaONPs were not toxic to macrophages; when these nanoparticles were used in combination with the *C. colocynthis* fruit extracts, they showed great cell viability. Cell viability was discovered to be dose dependent. The results of the current study’s findings are identical to those of previous studies by Alayed et al. [56] and de Souza et al. [57]. The in vitro hemolytic analysis of the synergistic solution of CaONPs and *C. colocynthis* fruit extracts showed only 16 percent hemolysis; the results make them fit for use in vivo. The findings of our study are in accordance with previously reported research by Eram et al. [58]. Nanoparticles show better bioactivities and safety when used synergistically with plant extracts due to the fabrication of bioactive phytochemicals on their surface [59,60,61]. In this study, we have also found that synergistic solution of CaONPs and CCFE show less toxicity and enhanced bioactivity. It might be due to the formation of chemical conjugates between nanoparticles and bioactive phytochemicals in CCFE. There are a large number of phenolics and flavonoids including syringic acid, quercetin, vanilic acid, sinapic acid, and feralic acid in CCFE [48], which have ability to cap and stabilize the nanoparticles, making them less toxic [62].

The antimicrobial activity of the synergistic solution of CaONPs and *C. colocynthis* fruit extracts was found to be significant. The antibacterial activity shown by *C. colocynthis* fruit extracts (CCFE) is due to presence of cucurbitacins, phenolic compounds, and flavonoids [63]. The antibacterial activity of greenly synthesized CaONPs is also due to the adsorption of flavonoids. Surprisingly, the synergistic solution of CaONPs and CCFE showed higher antibacterial potential. It might be due to the rupture of the cell wall due to the nanosize of CaONPs and potential protein inhibition by CCFE particularly due to cucurbitacins present in it. The activity index of the synergistic solution was slightly greater than that of streptomycin. It was depicted that CaONPs and *C. colocynthis* fruit extracts also possess antimicrobial potential, but when these are used synergistically, their antimicrobial potential is greatly increased. Our findings are consistent with those previously reported by Ahmad et al. [64] and Ikram et al. [65]. The assessment of antioxidant potential showed that CaONPs and *C. colocynthis* fruit extracts, as well as a synergistic solution of both CaONPs and *C. colocynthis* fruit extracts, were effective antioxidant agents. Antioxidant activity of CCFE may be attributed to presence of a large number of antioxidants including quercetin, kamferol, isovitexin, α-tocopherol, catechin, caffeic acid, ferulic acid, and gallic acid [66]. As the CaONPs have been reduced by the CCFE, some of the antioxidants—especially phenolic acids—have been deposited on the surface, due to which these NPs show antioxidant activity. The antioxidant activity was dose-dependent, but there was a significant difference in the antioxidant potential of the standard (BHT) and CaONPs. The results of our research are in line with the previously reported studies by Kandiah et al. [67].

Green synthesis of calcium oxide nanoparticles (CaONPs) from *C. colocynthis* fruit extracts (CCFE) presented in this research is a cheap, ecofriendly, and efficient method. CaONPs of nanosize with great surface area can be produced with ease by the method presented. The present study has found that greenly synthesized CaONPs have adsorbed several phytochemicals, including phenolics, carboxylic acids, alkanes, alkynes, amines, flavones, flavonols, and flavonoids. CaONPs have demonstrated excellent stability, biosafety, bioavailability, and bioactivities as a result of phytochemical adsorption. The synthesized CaONPs in this study have been investigated for in vivo–in vitro correlation and are found safe for use in vivo. These CaONPs can be used for studying in vivo bioactivities attributed to *C. colocynthis.* These NPs will improve drug delivery and increase the bioavailability of phytoconstituents adsorbed on them due to their small size, large surface area, and high adsorption of bioactive phytochemicals from CCFE.

## 5. Conclusions

It can be concluded that the green synthesis of calcium oxide nanoparticles is an easy and affordable process that can yield highly stable metallic nanoparticles. Greenly synthesized nanoparticles are not toxic to erythrocytes and possess negligible cytotoxicity. These nanoparticles possess significant antimicrobial and antioxidant activity. The fabrication of phenolics and flavonoids increased stability, making CaONPs suitable for use in vivo. More stabilizing procedures and using different approaches for the production of calcium oxide nanoparticles can pave the way for their pharmaceutical applications.

## Figures and Tables

**Figure 1 materials-16-02768-f001:**
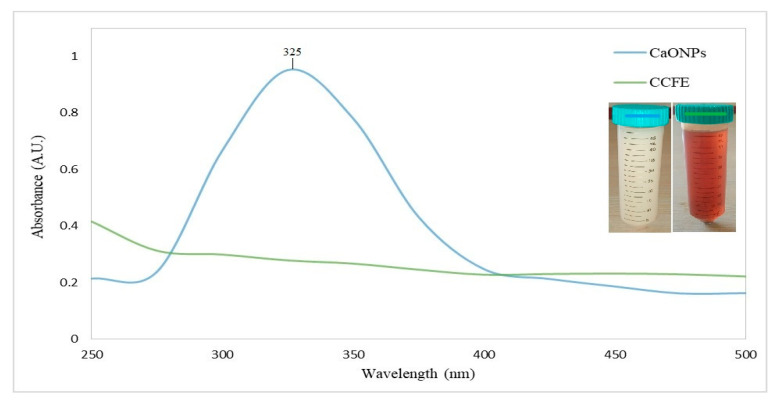
The UV-Vis spectrograph of calcium oxide nanoparticles. CaONPs—calcium oxide nanoparticles; CCFE—*C. colocynthis* fruit extracts.

**Figure 2 materials-16-02768-f002:**
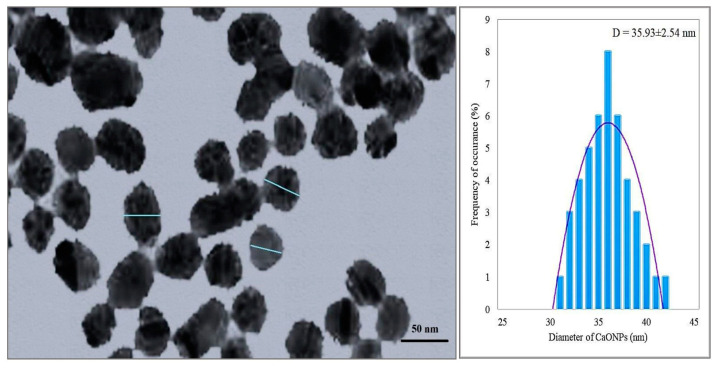
The transmission electron micrograph (TEM) of calcium oxide nanoparticles. D—Diameter of calcium oxide nanoparticles expressed as mean ± SEM.

**Figure 3 materials-16-02768-f003:**
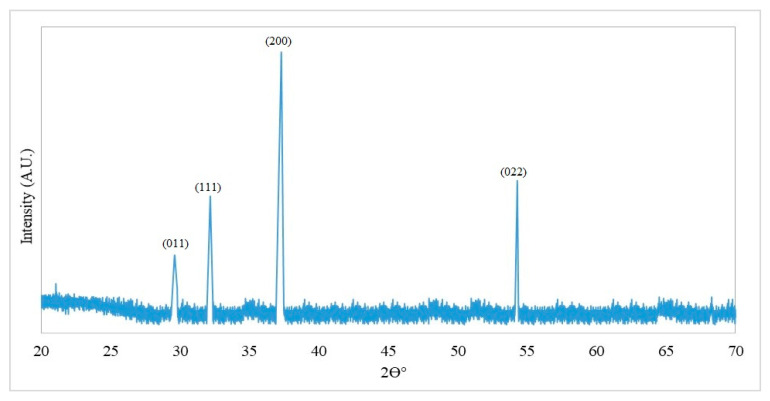
The X-ray diffraction (XRD) spectrograph of calcium oxide nanoparticles.

**Figure 4 materials-16-02768-f004:**
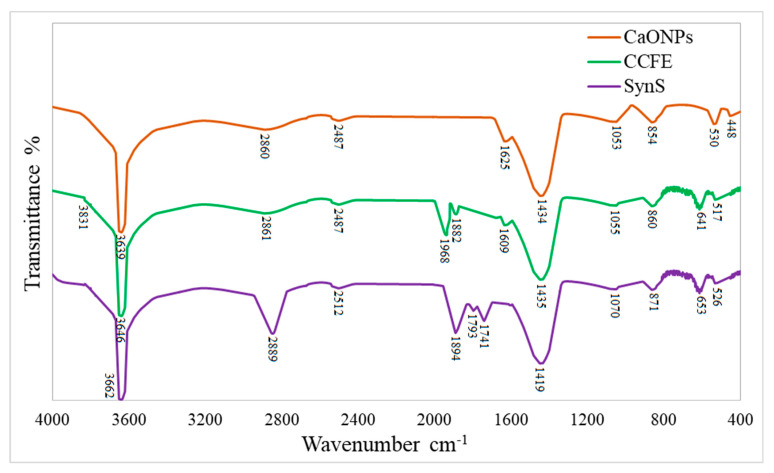
The Fourier-transform infrared (FTIR) spectra of calcium oxide nanoparticles. CaONPs—calcium oxide nanoparticles; CCFE—*C. colocynthis* fruit extracts, SynS—synergistic solution of CaONPs and CCFE.

**Figure 5 materials-16-02768-f005:**
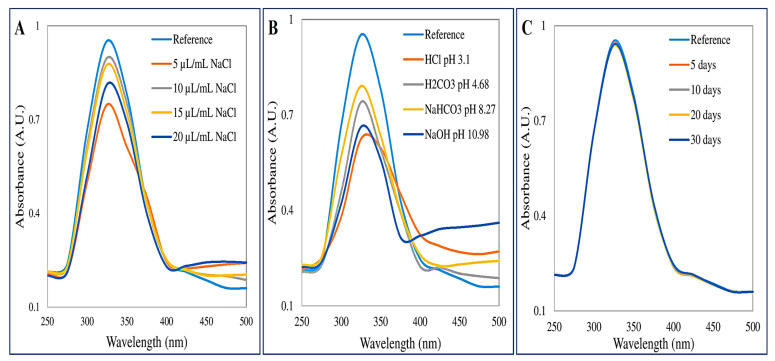
Stability analysis of calcium oxide nanoparticles with varying salinity (**A**); varying pH (**B**); passage of time (**C**).

**Figure 6 materials-16-02768-f006:**
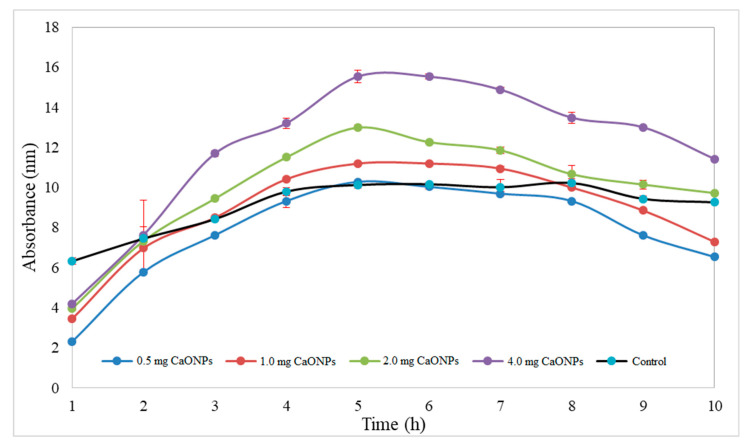
In vitro release of calcium oxide nonoparticles. Data are expressed as mean ± Standard error of means (n = 3).

**Figure 7 materials-16-02768-f007:**
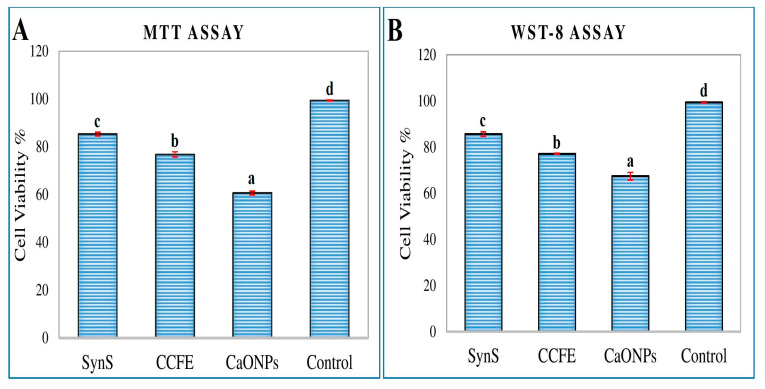
Cell viability percentages of CaONPs, *C. colocynthis* fruit extracts, and their synergistic solution by MTT assay (**A**) and WST-8 assay (**B**). SynS—a synergistic solution of calcium oxide nanoparticles and *C. colocynthis* fruit extracts; CCFE—*C. colocynthis* fruit extracts; CaONPs—calcium oxide nanoparticles. Data are expressed as mean ± SEM. In one-way ANOVA, the values *p* < 0.05 were considered significant. The Duncan Multiple Range Test (DMRT) was performed for values that were significantly different. The different alphabets on the bars indicate a significant difference in group means (n = 3). The same alphabets above the bars indicate there was no significant difference among the means of these groups.

**Figure 8 materials-16-02768-f008:**
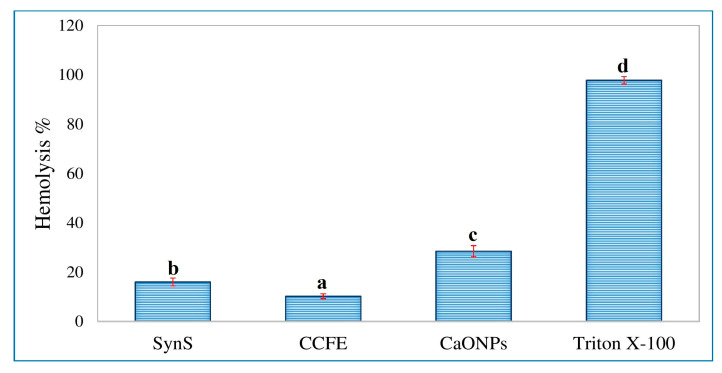
Hemolytic activity of CaONPs, *C. colocynthis* fruit extracts, and their synergistic solution. SynS—a synergistic solution of calcium oxide nanoparticles and *C. colocynthis* fruit extracts; CCFE—*C. colocynthis* fruit extracts; CaONPs—calcium oxide nanoparticles. Data are expressed as mean ± SEM. In one-way ANOVA, the values *p* < 0.05 were considered significant. The Duncan Multiple Range Test (DMRT) was performed for values that were significantly different. The different alphabets on the bars indicate a significant difference in group means (n = 3). The same alphabets above the bars indicate there was no significant difference among the means of these groups.

**Figure 9 materials-16-02768-f009:**
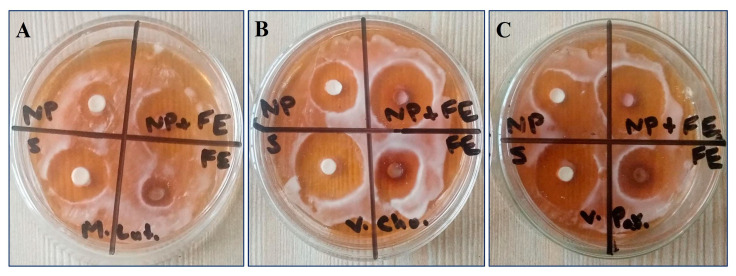
Antimicrobial activity of CaONPs, *C. colocynthis* fruit extracts and their synergistic solution against *Micrococcus luteus* (**A**), *Vibrio cholerae* (**B**), and *Vibrio parahaemolyticus* (**C**). NP—calcium oxide nanoparticles; FE—*C. colocynthis* fruit extracts; S—streptomycin; NP + FE—synergistic solution of calcium oxide nanoparticles and *C. colocynthis* fruit extracts.

**Table 1 materials-16-02768-t001:** Phytochemical analysis of *C. colocynthis* ethanolic fruit extracts.

Sr.	Phytochemical	Method	Point of Confirmation	Affirmation
01	Alkaloid	Dragendorff test	Formation of Red precipitates	+++
		Hager tests	Appearance of yellow color	+++
		Mayer test	Formation of yellow precipitates	+++
		Wagner test	Formation of reddish-brown precipitates	+++
02	Flavonoid	Alkaline reagent test	Disappearance of color	+
		Lead acetate test	Formation of yellow precipitates	+
		Magnesium ribbon test	Change in color from orange to red	++
03	Glycosides	Bromine water test	Appearance of brownish color	++
		Keller test	Formation of dark brown ring at surface	++
04	Phenols	Ferric chloride test	Appearance of bluish green color	+++
05	Saponins	Froth test	Formation of persistent froth	++
06	Tannins	Ferric chloride test	Appearance of blackish color	+
07	Terpenoids	Liebermann Burchard’s test	Formation of brown ring at surface	+++
		Salkowski’s test	Appearance of reddish color	+++

Key: + = present, ++ = present in good amount, +++ = present in excess amount.

**Table 2 materials-16-02768-t002:** Antimicrobial activity of CaONPs, *C. colocynthis* fruit extracts, and their synergistic solution.

Treatments	*Microccus luteus*		*Vibrio cholerae*		*Vibrio parahaemolyticus*	
	ZI	AI	ZI	AI	ZI	AI
CaONPs	7.3 ± 0.3 ^a^	0.29	11.8 ± 0.2 ^a^	0.39	10.4 ± 1.2 ^a^	0.4
CCFE	8.1 ± 0.5 ^b^	0.33	12.2 ± 0.6 ^a^	0.4	12.1 ± 0.4 ^b^	0.44
SynS	25.6 ± 1.1 ^d^	1.03	28.5 ± 0.3 ^b^	0.94	27.2 ± 1.3 ^c^	0.99
Streptomycin	24.8 ± 1.6 ^c^		30.2 ± 0.7 ^c^		27.4 ± 0.8 ^c^	

The values are expressed as the mean ± standard error of the means (n = 3). Different superscripts in a column indicate a significant (level of confidence, 95%) difference among the variables calculated by DMRT. CaONPs—calcium oxide nanoparticles; CCFE—*C. colocynthis* fruit extracts; SynS—a synergistic solution of calcium oxide nanoparticles and *C. colocynthis* fruit extracts; ZI—zone of inhibition; AI—activity index.

**Table 3 materials-16-02768-t003:** Antioxidant activity of CaONPs, *C. colocynthis* fruit extracts, and their synergistic solution, by DRSA assay.

Treatments	Concentration			
	250 µL/mL	500 µL/mL	1000 µL/mL	2000 µL/mL
CaONPs	31.97 ± 0.91 ^a^	35.55 ± 0.92 ^a^	41.19 ± 0.38 ^a^	49.88 ± 0.16 ^a^
CCFE	49.97 ± 1.09 ^b^	57.60 ± 0.59 ^b^	61.37 ± 1.19 ^b^	68.25 ± 0.37 ^b^
SynS	65.90 ± 0.97 ^c^	73.54 ± 0.69 ^c^	81.84 ± 3.14 ^c^	83.87 ± 1.68 ^c^
Control (BHT)	71.48 ± 0.09 ^d^	76.80 ± 2.72 ^d^	84.74 ± 4.09 ^d^	89.42 ± 4.19 ^d^

The values are expressed as the mean ± standard error of the means (n = 3). Different superscripts in a column indicate a significant (level of confidence, 95%) difference among the variables calculated by DMRT. In one-way ANOVA, the values *p* < 0.05 were considered significant. The Duncan Multiple Range Test (DMRT) was performed for values that were significantly different. The different alphabets on the bars indicate a significant difference in group means (n = 3). The same alphabets above the bars indicate there was no significant difference among the means of these groups. CaONPs—calcium oxide nanoparticles; CCFE—*C. colocynthis* fruit extracts; SynS—a synergistic solution of calcium oxide nanoparticles and *C. colocynthis* fruit extracts.

## Data Availability

All data are available on in this publication.

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
