# Peer review of "Biosynthesis and Characterization of Calcium Oxide Nanoparticles from Citrullus colocynthis Fruit Extracts; Their Biocompatibility and Bioactivities"

_materials, 2023, doi:10.3390/ma16072768_

Round 1
Reviewer 1 Report
The manuscript reported by Mazher et al. describes the fabrication of nanoparticles from Citrullus colocynthis fruit extracts and their characterizations. Variable methods have been employed for the characterizations, hence the manuscript provide not a few information for future studies following the current one. Therefore, I feel that this manuscript is potentially acceptable for the current journal after my minor concerns are all cleared.
-- I request the authors to care about the significant digits. For example, the size of the nanoparticles determined by TEM is described as "35.93 +/- 2.54 nm." However, it should not have the reliable accuracy of the order of 10^-2 nm. Do review any values shown in the main text as well as in Tables and Figures, confirm the digits, and revise if necessary.
-- Image quality of Figure 2 is too poor. It seems that the TEM micrograph has been extended vertically. Therefore, the nanoparticles look elongated improperly. Amend it.
-- I couldn't understand why the in vitro release of CaONPs has been examined. The text may state that this is because nanoparticles are studied for future use in living organisms in general. Then, how exactly the authors wants to apply CaONPs for use in living organisms? There can be a lot of ways of using nanoparitcles. Please describe introductory texts about the application that the authors expect for CaONPs using specific examples.
-- What is the "synergistic solution"? Is it just the mixture of CaONPs and C. colocynthis fruit extracts? If this is the mixture, what is the mixing ratio?
-- Cytotoxicity and hemolytic activity assays are confusing. For example, SynS demonstrates less cytotoxicity than CaONPs or CCFE. Why? If SynS is the mixture of CaONPs and CCFE, the cytotoxicity of SynS might be higher than that of only CaONPs or CCFE. Please discuss the possible reason for the mechanism of this interesting phenomenon
Author Response
- I request the authors to care about the significant digits. For example, the size of the nanoparticles determined by TEM is described as "35.93±2.54 nm." However, it should not have the reliable accuracy of the order of 10-2 Do review any values shown in the main text as well as in Tables and Figures, confirm the digits, and revise if necessary.
Response: We have carefully reviewed the significant digits as suggested.
- Image quality of Figure 2 is too poor. It seems that the TEM micrograph has been extended vertically. Therefore, the nanoparticles look elongated improperly. Amend it.
Response: The vertical expansion has been reduced and image has been improved as suggested.
- I couldn't understand why the in vitro release of CaONPs has been examined. The text may state that this is because nanoparticles are studied for future use in living organisms in general. Then, how exactly the authors wants to apply CaONPs for use in living organisms? There can be a lot of ways of using nanoparitcles. Please describe introductory texts about the application that the authors expect for CaONPs using specific examples.
Response: The in vitro release has been studied to formulate in vitro-in vivo correlation (IVIVC). The main hindrance in use of nanoparticles in living organisms is that they form coagulates and are extremely reactive in nature, we have studied the in vitro release of CaONPs to see if they have been properly capped and stabilized for in vivo use. Using dialysis membrane to study in vitro release kinetics is an easy and accurate method [D’Souza2014].
D’Souza, S. (2014). A Review of In Vitro Drug Release Test Methods for Nano-Sized Dosage Forms. Advances in pharmaceutics, 2014, 1-12.
The text added in MS “Citrullus colocynthis fruits are traditionally used for antidiabetic activity, modern research has also validated its use as antidiabetic activity due to presence of phenolics, flavonoids, and cucurbitacins in its extracts [Afzal et al. 2023; Drissi et al. 2021]. Calcium oxide nanoparticles have great ability of catalysis and drug delivery [Kouzu et al. 2008; Sadeghi and Husseini, 2013]. Green synthesis of CaONPs using fruit extracts of C. colocynthis can be useful for producing nanoparticles capped and stabilized with phytochemicals and having better bioavailability for treating different ailments including diabetes. We have studied the in vitro release kinetics to depict that these nanoparticles have great bioavailability for up to 10 hours. Findings of our research suggest that CaONPs synthesized using C. colocynthis fruit extracts can be suitable candidate for studying in vivo antidiabetic activity in future”.
- Drissi, F., Lahfa, F., Gonzalez, T., Peiretti, F., Tanti, J. F., Haddad, M. & Govers, R. A Citrullus colocynthis fruit extract acutely enhances insulin-induced GLUT4 translocation and glucose uptake in adipocytes by increasing PKB phosphorylation. Journal of Ethnopharmacology, 2021, 270, 113772. https://doi.org/10.1016/j.jep.2020.113772
- Afzal, M., Khan, A. S., Zeshan, B., Riaz, M., Ejaz, U., Saleem, A. & Ahmed, N. Characterization of Bioactive Compounds and Novel Proteins Derived from Promising Source Citrullus colocynthis along with In-Vitro and In-Vivo Activities. Molecules, 2023, 28(4), 1743. https://doi.org/10.3390/molecules28041743
- Kouzu, M., Kasuno, T., Tajika, M., Sugimoto, Y., Yamanaka, S., & Hidaka, J. Calcium oxide as a solid base catalyst for transesterification of soybean oil and its application to biodiesel production. Fuel, 2008, 87(12), 2798-2806. https://doi.org/10.1016/j.fuel.2007.10.019
- Sadeghi, M., & Husseini, M. H. A novel method for the synthesis of CaO nanoparticle for the decomposition of sulfurous pollutant. Journal of Applied Chemical Research, 2013, 7(4), 39-49. https://dorl.net/dor/20.1001.1.20083815.2013.7.4.4.3
- What is the "synergistic solution"? Is it just the mixture of CaONPs and C. colocynthis fruit extracts? If this is the mixture, what is the mixing ratio?
Response: Yes synergistic solution is a mixture of CaONPs and CCFE, containing 0.5 mg of CCFE and 0.5 mg of CaONPs in 1 mL of ethanol.
- Cytotoxicity and hemolytic activity assays are confusing. For example, SynS demonstrates less cytotoxicity than CaONPs or CCFE. Why? If SynS is the mixture of CaONPs and CCFE, the cytotoxicity of SynS might be higher than that of only CaONPs or CCFE. Please discuss the possible reason for the mechanism of this interesting phenomenon.
Response: Nanoparticles show better bioactivities and safety when used synergistically with plant extracts due to fabrication of bioactive phytochemicals on their surface [Barbinta-Patrascu et al. 2016; Essawy et al. 2021; Majeed et al. 2022]. In this study we have also found that synergistic solution of CaONPs and CCFE show less toxicity and enhanced bioactivity. It might be due to formation of chemical conjugates between nanoparticles and bioactive phytochemicals in CCFE. There are a large number of phenolics and flavonoids including syringic acid, quercetin, vanilic acid, sinapic acid and feralic acid in CCFE [Afzal et al. 2023], that have ability to cap and stabilize the nanoparticles making them less toxic [Liu et al. 2018].
- Barbinta-Patrascu, M. E., Badea, N., Pirvu, C., Bacalum, M., Ungureanu, C., Nadejde, P. L. & Rau, I. Multifunctional soft hybrid bio-platforms based on nano-silver and natural compounds. Materials Science and Engineering: C, 2016, 69, 922-932. https://doi.org/10.1016/j.msec.2016.07.077
- Essawy, E., Abdelfattah, M. S., El-Matbouli, M., & Saleh, M. Synergistic effect of biosynthesized silver nanoparticles and natural phenolic compounds against drug-resistant fish pathogens and their cytotoxicity: an in vitro study. Marine drugs, 2021, 19(1), 22. https://doi.org/10.3390/md19010022
- Majeed, M., Hakeem, K. R., & Rehman, R. U. Synergistic effect of plant extract coupled silver nanoparticles in various therapeutic applications-present insights and bottlenecks. Chemosphere, 2022, 288, 132527. https://doi.org/10.1016/j.chemosphere.2021.132527
- Afzal, M., Khan, A. S., Zeshan, B., Riaz, M., Ejaz, U., Saleem, A. & Ahmed, N. Characterization of Bioactive Compounds and Novel Proteins Derived from Promising Source Citrullus colocynthis along with In-Vitro and In-Vivo Activities. Molecules, 2023, 28(4), 1743. https://doi.org/10.3390/molecules28041743
- Liu, Y. S., Chang, Y. C., & Chen, H. H. Silver nanoparticle biosynthesis by using phenolic acids in rice husk extract as reducing agents and dispersants. Journal of food and drug analysis, 2018, 26(2), 649-656. http://dx.doi.org/10.1016/j.jfda.2017.07.005

Reviewer 2 Report
The submitted manuscript needs corrections to have work of quality with a good scientific contribution. The observations to be made and improved are described below:
Abstract
In line 36 “showing an absorbance peak at 325 nm.” the authors need to explain what this spike means.
In lines 36 and 37 "The XRD pattern showed sharp peaks that illustrated the crystalline nature of triturate", the authors need to mention what preferential crystal orientation is.
In lines 38 and 39 "The FTIR analysis of the CaONPs showed a coating of phytochemicals on their surface, due to which they showed great stability” the authors have to explain or show evidence of what they have written.
1. Introduction
The authors have to explain the objective and contribution of the work, since it is not clear. They only mention that they formulated a material and characterized it, however they do not say with what intention they did it or where it is directed. For this same reason the information presented in the introduction is insufficient, therefore it must be substantially improved.
For in vitro testing
I do not understand why it is used macrophages in the testing, the authors have to explain why their material is evaluated with macrophages.
FTIR (Figure 4)
The authors do not show the spectrum of the mixture of CaONPs and CCFE. How they can explain possible interactions or not of the materials involved?
Figure 5 has to be improved, the quality is low.
3.4. Assessment of in vitro release of CaONPs
The authors present a sustained release study without a control or reference group. For this case, it is necessary to show the absorbance of its base material, which is the dialysis membrane, and the membranes with their extract (CCFE), this in order to correctly interpret the absorbance and release.
Discussion
In lines 469 to 471 "A UV-Vis spectrograph of CaONPs synthesized from C. colocynthis fruit extracts showed an absorbance peak at 325 nm. Results of our research are the same as those reported for 332 nm by Gandhi et al. [34] and broad peaks in the range of 265 to 350 nm by Ramola et al.", the authors need to explain what this spike means.
The authors need to make a better discussion of the results, they have to explain the relationship of the results of the XRD pattern with the TEM micrograph, to consolidate the type of structure obtained and particle size, the discussion shown is weak and does not support what they explain.
For Fourier-transform infrared (FTIR) spectrum, the authors need to attend to the comment in figure 4 to make a better discussion in this section.
For cell viability and cytotoxicity, authors should explain why they used macrophages and explain their relationship to any application.
For the antimicrobial activity, the authors have to expand their discussion about how it is inhibiting bacterial growth, and an explanation of the effect of the material on the bacteria used; Just saying an antioxidant effect is not enough, it has to explain the action mechanism of the proposed material.

Author Response
|
Sr. |
Comment |
Response |
|
Abstract |
||
|
1 |
In line 36 “showing an absorbance peak at 325 nm.” The authors need to explain what this spike means |
Absorbance peak in this range depicts the coating of phenolic acids, flavones, flavonols and flavonoids on the surface of CaONPs. |
|
2 |
In lines 36 and 37 "The XRD pattern showed sharp peaks that illustrated the crystalline nature of triturate", the authors need to mention what preferential crystal orientation is. |
The XRD pattern illustrates preferred cubic crystalline nature of the triturate. |
|
3 |
In lines 38 and 39 "The FTIR analysis of the CaONPs showed a coating of phytochemicals on their surface, due to which they showed great stability” the authors have to explain or show evidence of what they have written. |
The vibrations present at 3639 cm-1 for alcohols or phenols, 2860 cm-1 for alkanes, 2487 cm-1 for alkynes, 1625 cm-1 for amines, and 1434 cm-1 for carboxylic acids show adsorption of phytochemicals on the surface of CaONPs. |
|
Introduction |
||
|
4 |
The authors have to explain the objective and contribution of the work, since it is not clear. They only mention that they formulated a material and characterized it, however they do not say with what intention they did it or where it is directed. For this same reason the information presented in the introduction is insufficient, therefore it must be substantially improved. |
This study presents a simple, ecofriendly, cost-effective, and easy method for the synthesis of calcium oxide nanoparticles (CaONPs) from C. colocynthis fruit extracts (CCFE) for the first time. It will elaborate on the physical and chemical nature of greenly synthesized CaONPs by UV-Vis spectroscopy, transmission electron microscopy (TEM), X-ray diffraction (XRD), and Fourier-transform infrared spectroscopy (FTIR). It will investigate in vitro-in vivo correlation (IVIVC) and the stability of greenly synthesized CaONPs for the first time. Greenly synthesized nanoparticles will be studied for biosafety by studying hemolytic activity and cytotoxicity assessment against macrophages. The antibacterial potential of CaONPs, CCFE, and their synergistic solution will be investigated against skin-borne pathogens, which has not been done before. Antioxidant potential will also be studied using different assays. Findings of this study will suggest the suitability and biocompatibility of greenly synthesized CaONPs for in vitro and in vivo bioactivities. |
|
For in vitro testing |
||
|
5 |
I do not understand why it is used macrophages in the testing, the authors have to explain why their material is evaluated with macrophages. |
Due to their nanosize, ability to form coagulations, and extremely reactive nature, nanoparticles can destroy cell membranes, which is a hindrance to their use in living organisms. We have studied the cytotoxic effect of greenly synthesized CaONPs against macrophages to assess their toxicity for living cells. Using macrophages to assess the cytotoxicity of plant extracts is an easy, cost-effective, and reliable method [de Oliveira et al. 2013; Naik et al. 2011]. The intention behind studying the biosafety of CaONPs is that, in the future, we want to study in vivo antidiabetic and antilipidemic activity using these nanoparticles. 1. de Oliveira, J. R., de Castro, V. C., Vilela, P. D. G. F., Camargo, S. E. A., Carvalho, C. A. T., Jorge, A. O. C., & de Oliveira, L. D. (2013). Cytotoxicity of Brazilian plant extracts against oral microorganisms of interest to dentistry. BMC complementary and alternative medicine, 13(1), 1-7. 2. Naik, S. K., Mohanty, S., Padhi, A., Pati, R., & Sonawane, A. (2014). Evaluation of antibacterial and cytotoxic activity of Artemisia nilagirica and Murraya koenigii leaf extracts against mycobacteria and macrophages. BMC complementary and alternative medicine, 14(1), 1-10. |
|
FTIR (Figure 4) |
||
|
6 |
The authors do not show the spectrum of the mixture of CaONPs and CCFE. How they can explain possible interactions or not of the materials involved? |
SynS is a mixture of both CaONPs and CCFE. The FTIR of SynS has also been added in Figure 4. It was noticed that the vibration for presence of alkanes was intensified in SynS and a new vibration of esters at 1741 cm‐1 was present. However, the strong vibrations for flavonoids, phenolics, and carboxylic acids were present in all test solutions including CCFE, CaONPs, and SynS. |
|
Figure 5 |
||
|
7 |
Figure 5 has to be improved, the quality is low. |
Figure 5 has been improved as suggested. (Attached at the end of this report). |
|
3.4. Assessment of in vitro release of CaONPs |
||
|
8 |
The authors present a sustained release study without a control or reference group. For this case, it is necessary to show the absorbance of its base material, which is the dialysis membrane, and the membranes with their extract (CCFE), this in order to correctly interpret the absorbance and release. |
The control has been added. Using dialysis membrane to study in vitro release kinetics is an easy and accurate method [D’Souza2014]. We have used metformin (Neodipar®, Sanofi, Pakistan) as a control drug in our in vitro release model. Because in the future we intend to use CaONPs prepared from CCFE for studying antidiabetic activity. 1. D’Souza, S. (2014). A Review of In Vitro Drug Release Test Methods for Nano-Sized Dosage Forms. Advances in pharmaceutics, 2014, 1-12. |
|
Discussion |
||
|
9 |
In lines 469 to 471 "A UV-Vis spectrograph of CaONPs synthesized from C. colocynthis fruit extracts showed an absorbance peak at 325 nm. Results of our research are the same as those reported for 332 nm by Gandhi et al.[34] and broad peaks in the range of 265 to 350 nm by Ramola et al.", the authors need to explain what this spike means. |
Absorbance peak in this range depicts the coating of hydroxicinnamic acids [Aleixandre-Tudo & Du Toit, 2018], flavones, flavonols and flavonoids [Gierschner et al. 2012] on the surface of CaONPs. Deposition of these phytochemicals have reduced the calcium ions into CaONPs. 1. Aleixandre-Tudo, J. L., & Du Toit, W. (2018). The role of UV-visible spectroscopy for phenolic compounds quantification in winemaking. Frontiers and new trends in the science of fermented food and beverages, 200-204. 2. Gierschner, J., Duroux, J. L., & Trouillas, P. (2012). UV/Visible spectra of natural polyphenols: A time-dependent density functional theory study. Food Chemistry, 131(1), 79-89. |
|
10 |
The authors need to make a better discussion of the results, they have to explain the relationship of the results of the XRD pattern with the TEM micrograph, to consolidate the type of structure obtained and particle size, the discussion shown is weak and does not support what they explain. |
A TEM micrograph showed a cubic to elliptical shape of CaONPs and an average size of 35.93±2.54 nm. Previously, CaONPs with average sizes ranging from 29 to 38 nm have been reported by Hussein et al. [38] and 13 to 49 nm by Mostafa et al. [39]. The XRD pattern of CaONPs depicted sharp diffractive peaks that showed the crystalline properties of triturate. The diffractive peaks were shown at 29.61° (011), 32.17° (111), 37.27° (200), and 54.26° (022), corresponding to CaO depicting the preferred cubic crystalline nature and average size of 32.12 nm calculated by following the Scherrer equation. ImageJ software calculated nanoparticle sizes from TEM ranging from 32 to 44 nm, with an average size of 35.93 nm. The XRD peaks were well matched with Joint Committee for Powder Diffraction Standard (JCPDS) Card No. 00‐004‐0777 also reported by Jadhav et al. [9]. |
|
11 |
For Fourier-transform infrared (FTIR) spectrum, the authors need to attend to the comment in figure 4 to make a better discussion in this section. |
The vibrations at 3639 cm-1 for -OH, 2860 cm-1 for -C-H), 2487 cm-1 for -C≡C-, 1625 cm-1 for -N-H, and 1434 cm-1 for -COOH show presence of free alcohols or phenols, alkanes, alkynes, amines, carboxylic acids, and aldehydes adsorbed on the surface of CaONPs. It was noticed that the vibration for presence of alkanes was intensified in SynS and a new vibration of esters at 1741 cm‐1 was present. However, the strong vibrations for flavonoids, phenolics, and carboxylic acids were present in all test solutions including CCFE, CaONPs, and SynS. |
|
12 |
For cell viability and cytotoxicity, authors should explain why they used macrophages and explain their relationship to any application. |
We have studied the cytotoxic effect of greenly synthesized CaONPs against macrophages to assess their toxicity for living cells. Intention for studying biosafety of CaONPs is that in future we want to study in vivo antidiabetic and antilipidemic activity using these nanoparticles. |
|
13 |
For the antimicrobial activity, the authors have to expand their discussion about how it is inhibiting bacterial growth, and an explanation of the effect of the material on the bacteria used. |
The antibacterial activity shown by C. colocynthis fruit extracts (CCFE) is due to presence of cucurbitacins, phenolic compounds and flavonoids [Rao & Poonia, 2023]. The antibacterial activity of greenly synthesized CaONPs is also due to adsorption of flavonoids. Surprisingly the synergistic solution of CaONPs and CCFE showed higher antibacterial potential. It might be due to rupture of cell wall due to nanosize of CaONPs and potential protein inhibition by CCFE particularly due to cucurbitacins present in it. 1. Rao, V., & Poonia, A. (2023). Citrullus colocynthis (bitter apple): bioactive compounds, nutritional profile, nutraceutical properties and potential food applications: a review. Food Production, Processing and Nutrition, 5(1), 4. |
|
14 |
Just saying an antioxidant effect is not enough, it has to explain the action mechanism of the proposed material. |
Antioxidant activity of CCFE may be attributed to presence of a large number of antioxidants including quercetin, kamferol, isovitexin, α‐tocopherol, catechin, caffeic acid, ferulic acid, and gallic acid [Khatri et al. 2021]. As the CaONPs have been reduced by the CCFE some of the antioxidants especially phenolic acids have been deposited on the surface due to which these NPs show antioxidant activity. 1. Khatri, S., Faizi, S., Fayyaz, S., & Iqbal, E. (2021). Citrullus colocynthis: A Treasure of Phytochemical, Pharmacological, Pesticidal and Nematicidal Compounds. Pakistan Journal of Nematology, 39(2). |

Reviewer 3 Report
The manuscript “Biosynthesis and characterization of calcium oxide nanoparticles from Citrullus colocynthis fruit extracts; their biocompatibility and bioactivities”, has interesting information about the greenly synthesis calcium nanoparticles using ethanolic fruit extracts of C. colocynthis. In addition, the characteristics and bioactivities of these nanoparticles were studied. The manuscript needs attention in some sections to improve the actual version. The comments are listed below.
1.- Introduction. This section must be improved, the relevance of the topic is missing. There is enough information about the synthesis of nanoparticles with plant extract, included its characterization. Then, it is not clear the innovation of this investigation.
2.- Methods. In Ethics declaration, how many Winter albino male rats were used?
3.- Discussion. This section must be improved, it is necessary to highlight the contribution of the results in the nanomaterials field. The authors only compared their results with others previously reported, but what is the contribution of this study?
Line 464-465, the authors wrote “The present research was focused on greenly synthesizing copper oxide nanoparticles (CaONPs) using C. colocynthis fruit extracts”, Is it correct?
4. Figures. In Figure 7, Does viability percentage correspond to the Y-axis?, this is not indicated in the Figures A and B. Same case in figure 8, it is not indicated the Y-Axis, is hemolytic activity?
Author Response
- This section must be improved, the relevance of the topic is missing. There is enough information about the synthesis of nanoparticles with plant extract, included its characterization. Then, it is not clear the innovation of this investigation.
Response: C. colocynthis is rich in cucurbitacins [Kim et al. 2018], flavonoids [Marzouk et al. 2022], phenolics and natural antioxidants [Al-Nablsi et al. 2022]. Plant secondary metabolites are helpful to reduce, cap and stabilize metallic nanoparticles [Nagore et al. 2021], phenolics and flavonoids are most effective biological reducers [Roy et al. 2022]. Calcium oxide is biocompatible and possess several biomedical applications specifically antibacterial potential [Nipunika et al. 2022]. Green synthesis of calcium oxide nanoparticles from organic molecules is cost-effective and easy method [Hu et al. 2015].
This study presents a simple, ecofriendly, cost-effective, and easy method for the synthesis of calcium oxide nanoparticles (CaONPs) from C. colocynthis fruit extracts (CCFE) for the first time. It will elaborate on the physical and chemical nature of greenly synthesized CaONPs by UV-Vis spectroscopy, transmission electron microscopy (TEM), X-ray diffraction (XRD), and Fourier-transform infrared spectroscopy (FTIR). It will investigate in vitro-in vivo correlation (IVIVC) and the stability of greenly synthesized CaONPs for the first time. Greenly synthesized nanoparticles will be studied for biosafety by studying hemolytic activity and cytotoxicity assessment against macrophages. The antibacterial potential of CaONPs, CCFE, and their synergistic solution will be investigated against skin-borne pathogens, which has not been done before. Antioxidant potential will also be studied using different assays. Findings of this study will suggest the suitability and biocompatibility of greenly synthesized CaONPs for in vitro and in vivo bioactivities.
- Kim, Y. C., Choi, D., Zhang, C., Liu, H. F., & Lee, S. (2018). Profiling cucurbitacins from diverse watermelons (Citrullus spp.). Horticulture, Environment, and Biotechnology, 59, 557-566. https://doi.org/10.1007/s13580-018-0066-3
- Al-Nablsi, S., El-Keblawy, A., Ali, M. A., Mosa, K. A., Hamoda, A. M., Shanableh, A. & Soliman, S. S. (2022). Phenolic contents and antioxidant activity of Citrullus Colocynthis fruits, growing in the hot arid desert of the UAE, influenced by the fruit parts, accessions, and seasons of fruit collection. Antioxidants, 11(4), 656. https://doi.org/10.3390/antiox11040656
- Marzouk, B., Refifà, M., Montalbano, S., Buschini, A., Negri, S., Commisso, M., & Degola, F. (2022). In Vitro Sprouted Plantlets of Citrullus colocynthis (L.) Schrad Shown to Possess Interesting Levels of Cucurbitacins and Other Bioactives against Pathogenic Fungi. Plants, 11(20), 2711. https://doi.org/10.3390/plants11202711
- Nagore, P., Ghotekar, S., Mane, K., Ghoti, A., Bilal, M., & Roy, A. (2021). Structural properties and antimicrobial activities of Polyalthia longifolia leaf extract-mediated CuO nanoparticles. BioNanoScience, 11, 579-589. https://doi.org/10.1007/s12668-021-00851-4
- Roy, A., Khan, A., Ahmad, I., Alghamdi, S., Rajab, B. S., Babalghith, A. O. & Islam, M. (2022). Flavonoids a bioactive compound from medicinal plants and its therapeutic applications. BioMed Research International, 2022. https://doi.org/10.1155/2022/5445291
- Nipunika, U., Jayaneththi, Y., & Sewwandi, G. A. (2022, July). Synthesis of Calcium Oxide Nanoparticles from Waste Eggshells. In 2022 Moratuwa Engineering Research Conference (MERCon)(pp. 1-5). IEEE. https://doi.org/10.1109/MERCon55799.2022.9906264
- Hu, K., Wang, H., Liu, Y., & Yang, C. (2015). KNO3/CaO as cost-effective heterogeneous catalyst for the synthesis of glycerol carbonate from glycerol and dimethyl carbonate. Journal of Industrial and Engineering Chemistry, 28, 334-343. https://doi.org/10.1016/j.jiec.2015.03.012
- In Ethics declaration, how many Wister albino male rats were used?
Response: Three Wister albino male rats were used in this study.
- This section must be improved, it is necessary to highlight the contribution of the results in the nanomaterials field. The authors only compared their results with others previously reported, but what is the contribution of this study?
Response: Green synthesis of calcium oxide nanoparticles (CaONPs) from C. colocynthis fruit extracts (CCFE) presented in this research is a cheap, ecofriendly, and prompt method. CaONPs of nanosize with great surface area can be produced with ease by the method presented. The present study has found that greenly synthesized CaONPs have adsorbed several phytochemicals, including phenolics, carboxylic acids, alkanes, alkynes, amines, flavones, flavonols, and flavonoids. CaONPs have demonstrated excellent stability, biosafety, bioavailability, and bioactivities as a result of phytochemical adsorption. The synthesized CaONPs in this study have been investigated for in vivo-in vitro correlation and are found safe for use in vivo. These CaONPs can be used for studying in vivo bioactivities attributed to C. colocynthis. These NPs will improve drug delivery and increase the bioavailability of phytoconstituents adsorbed on them due to their small size, large surface area, and high adsorption of bioactive phytochemicals from CCFE.
Line 464-465, the authors wrote “The present research was focused on greenly synthesizing copper oxide nanoparticles (CaONPs) using C. colocynthis fruit extracts”, Is it correct?
Response: We are very sorry it was a mistake. We have corrected it.
- In Figure 7, does viability percentage correspond to the Y-axis? This is not indicated in the Figures A and B. Same case in figure 8, it is not indicated the Y-Axis, is hemolytic activity?
Response: Yes in Figure 7 viability percentage corresponds to Y-axis. Figures 7A and 7B have been amended. In Figure 8 Y-axis represents hemolytic activity. Figure 8 has also been amended.

Round 2
Reviewer 2 Report
The authors made their comments correctly. There are details but with what was corrected the quality of the manuscript improved substantially.
Reviewer 3 Report
The manuscript was improved according to reviewer's comments.